# DyMixOp: Guiding Neural Operator Design for PDEs from a Complex Dynamics Perspective with Local-Global-Mixing

## Abstract

A primary challenge in using neural networks to approximate nonlinear dynamical systems governed by partial differential equations (PDEs) is transforming these systems into a suitable format, especially when dealing with non-linearizable dynamics or the need for infinite-dimensional spaces for linearization. This paper introduces DyMixOp, a novel neural operator framework for PDEs that integrates insights from complex dynamical systems to address this challenge. Grounded in inertial manifold theory, DyMixOp transforms infinite-dimensional nonlinear PDE dynamics into a finite-dimensional latent space, establishing a structured foundation that maintains essential nonlinear interactions and enhances physical interpretability. A key innovation is the Local-Global-Mixing (LGM) transformation, inspired by convection dynamics in turbulence. This transformation effectively captures both fine-scale details and nonlinear interactions, while mitigating spectral bias commonly found in existing neural operators. The framework is further strengthened by a dynamics-informed architecture that connects multiple LGM layers to approximate linear and nonlinear dynamics, reflecting the temporal evolution of dynamical systems. Experimental results across diverse PDE benchmarks demonstrate that DyMixOp achieves state-of-the-art performance, significantly reducing prediction errors, particularly in convection-dominated scenarios reaching up to 86.7%, while maintaining computational efficiency and scalability.

## 1 Introduction

Partial differential equations (PDEs) are the mathematical backbone for describing chaotic behaviors and understanding underlying mechanics in dynamical systems. They are distributed in various fields including climate (Bi et al., 2023; Lam et al., 2023), molecular dynamics (Rapaport, 2004), ecological modeling (Blasius et al., 1999), brain activity (Breakspear, 2017), chemistry (Jensen, 2017), heat transfer (Howell et al., 2020) and turbulent flows (Mukherjee et al., 2023). Predicting the dynamics of complex systems by solving PDEs is crucial for scientific and engineering applications. As a result, a variety of numerical methods have been developed (Dennis Jr & Schnabel, 1996; Moin & Mahesh, 1998), including the finite difference method (Smith, 1985), finite volume method (Versteeg, 2007), lattice Boltzmann method (Succi, 2001) and finite element method (Zienkiewicz, 1971). Despite the high fidelity of simulations produced by traditional approaches, they become inefficient with frequent recalculations whenever initial conditions or equation parameters are altered.

In recent years, data-driven methods have thrived across various disciplines (Jordan & Mitchell, 2015). Among these approaches, neural networks have garnered significant attention from researchers due to their excellent performance in other fields such as large language modeling (Bahdanau, 2014), image recognition (Simonyan & Zisserman, 2014) and games (Mnih et al., 2015; Vinyals et al., 2019). Therefore they are perceived as promising tools to overcome the limitations of traditional methods and offer more possibilities for solving ill-defined problems. In terms of the approximation way whether the transformation of the neural layer spans the entire input domain, they can be roughly classified into four kinds: **the local transformation**, **the global transformation**, **the local-global adding transformation (LGA)**, **the local-local mixing transformation (LLM)**. Due to the complete architecture complexity of neural networks, here we only focus on the single

neural layer. The general transformation of a neural layer can be defined in the form of

$$(G_\theta v)(x) = \int_{D_\tau} g_\theta(x, \tau) v(\tau) d\tau, \quad x \in D_x \tag{1}$$

where $G_\theta$ is a integral transformation parameterized by parameter sets $\theta \in \Theta$ in neural networks, $g_\theta$ represents a integral kernel function parameterized by neural networks, $D_\tau$ and $D_x$ are bounded domains in $\mathbb{R}^d$ ($d \in \mathbb{Z}$ denotes the spatial dimension), $\tau \in D_\tau$ and $x \in D_x$ are variables in the input and output domains respectively, and $v$ is defined on $D_\tau$. When $D_\tau$ is the partial domain $P_\tau \subset \mathbb{R}^d$, it is the local transformation. When $D_\tau$ is the entire domain $E_\tau \subset \mathbb{R}^d$, it is the global transformation. Mixing here means the element-wise product and adding means the element-wise addition between transformations.

## 2 RELATED WORK

**Models with Local Transformations**. In the 1990s, Lagaris et al. (1998) trained a shallow FCNN to solve PDEs. With the recent revival of deep learning (LeCun et al., 2015), Raissi et al. (2019) developed this idea based on modern techniques of deep learning and proposed physics-informed neural networks (PINNs) to solve PDEs. Many efforts have been made to develop PINNs (Pang et al., 2019; Lu et al., 2021b; Karniadakis et al., 2021). To avoid unaffordable computational cost, all of the above studies limited input to only a few points at once and thus are the local transformation. To address the curse of dimensionality in image inputs, plenty of works were performed to approximate the evolution operator using CNNs (Qu et al., 2022; Gao et al., 2021; List et al., 2022). Some convolutional neural layer-based network architectures were also developed to solve PDE problems, including generative adversarial networks for two-dimensional turbulence (Kim et al., 2024; 2021), and autoencoder for three-dimensional turbulence (Xuan & Shen, 2023). Obviously, these convolutional neural layer-based network architectures fall into the scope of the local transformation.

**Models with Global Transformations**. The self-attention blocks in Transformers model using the composite matricial local-local mixing transformations belong to the global transformation (Vaswani, 2017), and how the kernel function can be extended to the vanilla transformer formula is detailed in (Kovachki et al., 2023). Some researches were performed to construct a transformer-based framework to solve PDE problems (Cao, 2021; Li et al., 2022).

**Models with LLM Transformations**. More complex neural network architectures targeted at LLM convolutional transformations for a more powerful nonlinear representation. Shi et al. (2015) combined convolutional transformations with the classical long short-term memory networks (LSTMs), where the cell state and latent state of LSTMs are obtained from element-wise products. Long et al. (2018) combined the traditional numerical schemes with the LLM convolutional transformation to solve PDEs problems. Similarly, Rao et al. (2023) also developed a framework to encode the physics and numerical schemes based on LLM convolutional transformation. However, the local convolutional transformation heavily depends on mesh discretization, typically leading to a degraded performance under changes in mesh discretization.

**Models with LGA Transformations**. To address mesh dependency in local convolutional transformations, two approaches exist: constructing networks with mesh-independent discrete methods, such as one-size convolutional kernels, or approximating the kernel in an auxiliary space. The graph kernel network framework approximates the kernel in graph space using local convolutional and graph transformations (Li et al., 2020b). To incorporate global information and improve efficiency, spectral methods were introduced, representing complex patterns compactly (Trefethen, 2000). Li et al. (2020a) developed the Fourier neural operator, replacing graph approximations with Fourier layers to process entire domains. This inspired other spectral-based operators to transform kernels into spectral spaces, such as wavelet (Tripura & Chakraborty, 2022) and Laplace neural operators (Cao et al., 2024).

**Other Models**. It is worth mentioning that due to the variety of network variants and high-level defined architectures, it is sometimes hard to directly classify their transformation. For example, the concept of the neural operator was also presented by Lu et al. (2021a). . They developed the DeepONet framework composed of the trunk network and branch network but transformations are not explicitly defined.

## 3 CHALLENGES AND METHODOLOGIES

**Operator Definition**. In this work, we focus on the form of neural operators due to their mesh independence and generalization to unseen parameters. Here, we first outline the concept of operator approximation. Given a spatial domain $D \subset \mathbb{R}^d$ with a spatial dimension $d$, an operator mapping from the input space to the solution space is defined as:

$$\mathcal{G} : \mathcal{I}(D; \mathbb{R}^{d_i}) \to \mathcal{U}(D; \mathbb{R}^{d_u}), \tag{2}$$

where $\mathcal{I}(D; \mathbb{R}^{d_i})$ and $\mathcal{U}(D; \mathbb{R}^{d_u})$ are Banach spaces representing the input and solution spaces, respectively, and $d_i, d_u \in \mathbb{N}$ denote their corresponding dimensions. Specifically, it often involves the evolution of the system in the context of PDEs and thus considers $\mathcal{G}$ an evolutionary operator.

**Challenges in Nonlinear Dynamical System Approximation**. To design a neural evolution operator that can effectively approximate autonomous dynamical systems, we first need to understand the intrinsic connection between the original dynamical system and the dynamics expressed within the neural network framework. This theoretical foundation is crucial for guiding us on how to simplify the infinite-dimensional partial differential equation dynamics into a finite-dimensional and expressive latent dynamics. The general form of an autonomous dynamical system is:

$$\frac{\partial u(t)}{\partial t} = F(u), \ t \in [0, T] \tag{3}$$

where $u \in \mathcal{U}(D; \mathbb{R}^{d_u})$ is a solution of dynamical systems, and $F : \mathcal{U}(D; \mathbb{R}^{d_u}) \to \mathcal{U}(D; \mathbb{R}^{d_u})$ is an operator that acts on the solution $u$ at each time $t$ and represents the dynamics of $u$. The operator $F$ may consist of linear operators, nonlinear operators, and source terms, depending on the specific dynamical system being modeled. Generally, nonlinear dynamical systems are addressed by transforming the system into a (partially) linearized form or mapping it to a new representation suitable for analysis and control utilizing a dimension-shifting operator $\mathscr{T}$. After applying $\mathscr{T} : \mathcal{U}(D; \mathbb{R}^{d_u}) \to \mathcal{V}(D; \mathbb{R}^{d_v})$, the solution $u$ is transformed to a latent state $v = \mathscr{T}(u)$ in a latent space $\mathcal{V}(D; \mathbb{R}^{d_v})$, with its latent dynamics given by:

$$\frac{\partial v(t)}{\partial t} = \tilde{F}(v), \quad t \in [0, T] \tag{4}$$

where $\tilde{F} : \mathcal{V}(D; \mathbb{R}^{d_v}) \to \mathcal{V}(D; \mathbb{R}^{d_v})$ is an operator that acts on $v$ and represents the dynamics of $v$. Unfortunately, for known dynamical systems, the operator $\mathscr{T}$ is difficult to determine; for unknown dynamical systems, even $F(u)$ remains unresolved. These difficulties pose significant challenges for solving arbitrary complex dynamical systems. Hopefully, machine learning methods can approximate them in aid of data alone but often implicitly assume locally linearized dynamical systems or globally linearizable nonlinear systems (Cenedese et al., 2022).

However, under this simplified assumption, two major challenges arise: (i) achieving linearization for nonlinear systems often requires an infinite-dimensional latent space $\mathcal{V}$ (Brunton et al., 2022), i.e. $d_v \to \infty$, making direct computation impossible, and (ii) some dynamical systems may possess intrinsically non-linearizable dynamics, meaning $\tilde{F}$ retains nonlinear components even in an infinite-dimensional $\mathcal{V}$ (Cenedese et al., 2022).

**Inertial Manifold Theory for Dimensional Reduction**. Fortunately, inertial manifold theory (Foias et al., 1988) provides a rigorous framework for (i) reducing such systems to finite-dimensional dynamics that capture the essential long-term behavior and (ii) remaining necessary nonlinearity. This approach offers a principled foundation for developing physically interpretable models.

An inertial manifold $\mathcal{M}$ is a finite-dimensional, Lipschitz continuous manifold embedded within a Hilbert space $\mathcal{H}$. Any state $v \in \mathcal{M}$ can be uniquely decomposed into a low-mode component $\hat{v}$ and a high-mode component that is a function of the low-mode one $v = \hat{v} + \Phi(\hat{v})$. Here, $\hat{v} = \mathscr{P}_m v$ is the projection of $v$ onto the first $d_m$ eigenmodes, and $\Phi$ is a Lipschitz continuous function mapping the low modes to the corresponding high modes in the orthogonal complement space.

Consider latent dynamics governed by $\frac{\partial v}{\partial t} = \mathcal{L}(v) + \mathcal{N}(v)$, where $\mathcal{L}$ and $\mathcal{N}$ are linear and nonlinear operators, respectively. If the long-term dynamics are confined to an inertial manifold $\mathcal{M}$, we can project the governing equations onto the low-mode space $\mathcal{V}_r = \mathscr{P}_m \mathcal{V}$. Substituting $v = \hat{v} + \Phi(\hat{v})$ into the dynamics and applying the projection $\mathscr{P}_m$ yields the exact reduced dynamics for $\hat{v}$:

$$\frac{\partial \hat{v}}{\partial t} = \mathscr{P}_m \mathcal{L}(\hat{v} + \Phi(\hat{v})) + \mathscr{P}_m \mathcal{N}(\hat{v} + \Phi(\hat{v}))$$

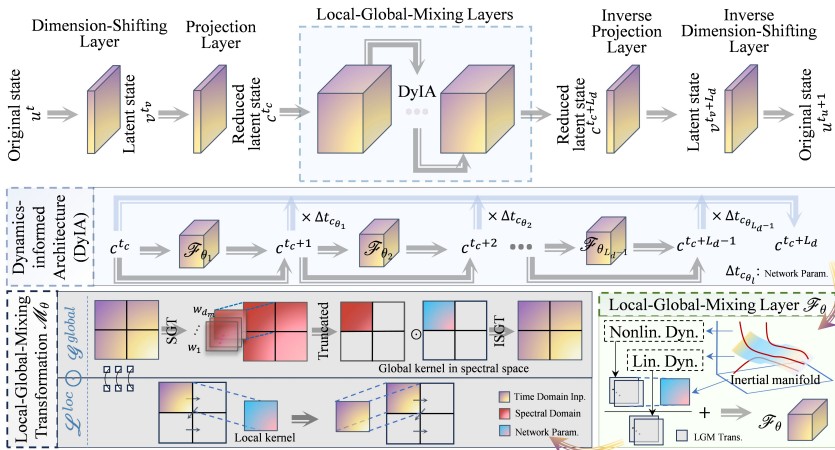

Figure 1: Illustration of the DyMixOp model composed of the dimension-shifting layer and its inverse, the projection layer and its inverse, and the LGM layers that adopt LGM transformations to approximate dynamics within the dynamics-informed architecture.

Under the common assumption that the nonlinear interactions involving the slave high modes $\Phi(\hat{v})$ can be modeled as a correction term, we can approximate the above as:

$$\frac{\partial \hat{v}}{\partial t} \approx \mathcal{L}\hat{v} + \mathscr{P}_m \mathcal{N}(\hat{v}) + \mathscr{P}_m \mathcal{R}[\mathcal{N}(\hat{v})] \tag{5}$$

where $\mathcal{R}$ is a residual operator capturing the influence of the high-mode interactions. This derivation offers a clear blueprint for constructing a reduced-order model and motivates the following design principle of our neural operator by translating the latent state $\hat{v}$ into the spectral coefficients $c$ (further derivation under additional assumptions see the Appendix).

**Proposition 1.** *(Principled Architectural Design) Our proposed neural operator, $\mathcal{G}^\dagger$, is constructed as a direct functional analogue of the reduced dynamics dictated by inertial manifold theory. By projecting Equation 5 onto a spectral basis to obtain dynamics for the reduced latent state $c$ (by acting $\mathscr{P}_m$ on $v$), the network is designed to approximate the evolution operator $\mathscr{F}(c)$ where*

$$\frac{\partial c(t)}{\partial t} = \mathscr{F}(c) \approx \mathcal{L}_c c + \mathcal{A}[\mathcal{N}_c(c)] \tag{6}$$

*Specifically, the architecture $\mathcal{G}^\dagger$ is structured to explicitly model the components of $\mathscr{F}(c)$: (i) a linear component approximates the linear operator $\mathcal{L}_c : \mathcal{C}(D; \mathbb{R}^{d_m}) \to \mathcal{C}(D; \mathbb{R}^{d_m})$, and (ii) a nonlinear component approximates the operator $\mathcal{A}[\mathcal{N}_c(\cdot)]$, where $\mathcal{N}_c : \mathcal{C}(D; \mathbb{R}^{d_m}) \to \mathcal{C}(D; \mathbb{R}^{d_m})$ is a nonlinear operator, $\mathcal{A} : \mathcal{C}(D; \mathbb{R}^{d_m}) \to \mathcal{C}(D; \mathbb{R}^{d_m})$ is an operator mapping the low-mode component to the full-mode component.*

This proposition formalizes the link between theory and method. Instead of being merely inspired by the theory, our network architecture is a direct implementation of the mathematical structure that the theory predicts. This provides a principled foundation for designing neural layers that approximate arbitrary complex dynamics with enhanced physical interpretability and generalization capabilities.

**Local-Global-Mixing Transformation**. Inspired by the profound impact of complex physical systems on network design (Ho et al., 2020; Chen et al., 2018), particularly the multi-scale dynamics of turbulence and the role of its convection term, we sought to create a transformation capable of capturing similarly intricate dynamics. The convection term, defined by the product of a variable $c$ and its gradient $\frac{\partial c}{\partial x}$, naturally embodies a form of local-global interaction. The variable $c$ itself is intrinsically localized, tied to its specific spatial position, suggesting the behavior of a local operation like convolution. In contrast, the computation of its gradient $\frac{\partial c}{\partial x}$ relies on information from the wider domain, often requiring global approaches such as spectral methods. This inherent structure within the convection term, where local state mixes with global gradient information, provides the core inspiration for our novel LGM transformation. This architecture is designed to explicitly combine the benefits of local and spectral global transformations, mirroring the essential mixing process

observed in convection to handle complex multi-scale features and mitigating the spectral bias in existing neural operators, as follows:

**Definition 1.** *(Local-Global-Mixing Transformation) The Local-Global-Mixing (LGM) transformation is defined as a parameterized operator that combines localized and global information from the input state $c$ in an element-wise multiplicative form. Specifically, it is given by:*

$$\mathscr{M}_\theta(c) = \mathscr{L}_\theta^{loc}(c) \odot \mathscr{G}_\theta^{glob}(c) = \left( \int_{P_\tau} p_\theta(x,\tau)c(\tau)d\tau \right) \odot \left( \int_{E_\tau} e_\theta(x,\tau)c(\tau)d\tau \right), \quad (7)$$

*where $\mathscr{L}_\theta^{loc}$ is the local integral operator, $\mathscr{G}_\theta^{glob}$ is the global integral operator, $c$ is the reduced latent state. $p_\theta(x,\tau): D_x \times P_\tau \to \mathbb{R}$ is the local kernel function, responsible for capturing fine-grained, position-specific interactions. $e_\theta(x,\tau): D_x \times E_\tau \to \mathbb{R}$ is the global kernel function, designed to incorporate domain-wide information and relationships. They are parameterized by a set of parameters $\theta$. $P_\tau$ and $E_\tau$ are integration domains corresponding to local and global interactions, respectively. $\odot$ denotes the element-wise (Hadamard) product.*

The transformation provides a versatile way to represent both linear and nonlinear dynamics. (i) For linear dynamics, setting $p_\theta(x,\tau) = \frac{1}{P_\tau c}$ or $e_\theta(x,\tau) = \frac{1}{E_\tau c}$ makes $\mathscr{M}_\theta$ a linear integral operator. (ii) For nonlinear dynamics, parameterized kernels $p_\theta(x,\tau)$ and $e_\theta(x,\tau)$ turn $\mathscr{M}_\theta$ into a nonlinear operator. This flexibility allows the LGM transformation to model both types effectively, offering a unified approach for complex dynamical systems.

**Local-Global-Mixing Layers**. Due to the flexibility of the LGM transformation in representing both linear and nonlinear dynamics, we employ two separate LGM transformations to approximate the linear component $\mathcal{L}_c$ and the nonlinear component $\mathcal{N}_c$ in the above design principle, with the reduced latent state as input.

$\mathcal{A}$ primarily involves high-mode small quantities that emphasize local interactions. So we can use a local transformation to approximate the operator $\mathcal{A}$, facilitating the complexity of neural layers, as follows:

$$\mathscr{F}_\theta(c) = \mathscr{M}_\theta^{\mathcal{L}}(c) + \mathscr{H}_\theta \circ \mathscr{M}_\theta^{\mathcal{N}}(c), \quad (8)$$

where $\mathscr{M}_\theta^{\mathcal{L}} : \mathcal{C}(D; \mathbb{R}^{d_m}) \to \mathcal{C}(D; \mathbb{R}^{d_m})$, $\mathscr{M}_\theta^{\mathcal{N}} : \mathcal{C}(D; \mathbb{R}^{d_m}) \to \mathcal{C}(D; \mathbb{R}^{d_m})$ indicate LGM transformations approximating the linear dynamics $\mathcal{L}_c c$ and the nonlinear dynamics $\mathcal{N}_c(c)$, respectively, and $\mathscr{H}_\theta : \mathcal{C}(D; \mathbb{R}^{d_m}) \to \mathcal{C}(D; \mathbb{R}^{d_m})$ the local transformation. Sometimes dynamical systems may involve composite operators, such as two higher-order derivatives in the Kuramoto-Sivashinsky system, and one can utilize more transformations to finely approximate different components in operators. Finally, we can construct a LGM layer, which considers the reduced latent state $c$ as input, to represent the reduced latent dynamics with the definition as follows:

**Definition 2.** *(Reduced Latent Dynamics Represented by Single LGM Layer) The reduced latent dynamics represented by a single Local-Global-Mixing (LGM) neural layer is a sum of a local transformation $\mathscr{H}_\theta$ and multiple LGM transformations $\mathscr{M}_\theta$, each capturing distinct components or scales of the system's behavior. Specifically, the dynamics are formulated as:*

$$\mathscr{F}_\theta(c) = \sum_{a=1}^{n_l} \mathscr{M}_\theta^{\mathcal{L}^a}(c) + \mathscr{H}_\theta \Big[ \sum_{b=1}^{n_n} \mathscr{M}_\theta^{\mathcal{N}^b}(c) \Big], \quad (9)$$

*where $n_l, n_n$ are the numbers of the linear and nonlinear LGM transformation, respectively, and $c$ is the reduced latent state.*

**Dynamics-Informed Architecture**. In deep learning, enhancing the representational capacity of neural networks is often achieved by stacking multiple layers. To integrate this concept into the DyMixOp framework, an architecture designed for connecting LGM layers is proposed based on the evolution of dynamics. By introducing time transformations between each dynamics and learnable evolutionary step (derivation see Appendix), we can get the following formula of the output:

$$c_{L_d} = c_0 + \sum_{l=1}^{L_d} \Delta t_{c_{\theta_l}} \mathscr{F}_{\theta_l}(c_{l-1}) \quad (10)$$

where $\Delta t_{c_{\theta_l}}$ is a parameterized evolutionary step and the input to each layer $c_{l-1}$ is recursively defined as $c_{l-1} = c_{l-2} + \mathscr{F}_{\theta_{l-1}}(c_{l-2}), l = 2, ..., L_d$.

**DyMixOp**. Here we can assemble the above components to propose a novel neural operator named DyMixOp shown in Fig. 1. For clarity, let us assume $k = 0$ and then denote the temporal input sequence $[u^{t_{i-k}}, \ldots, u^{t_i}]$ by $u^t$. Given a depth $L_d \in \mathbb{N}$, the DyMixOp takes the following composite form:

$$\mathcal{G}^{\dagger}(u^t; \theta) = \mathscr{T}^{-1} \circ \mathscr{P}_m^{-1} \circ \mathscr{C} \circ \mathscr{P}_m \circ \mathscr{T}(u^t), \tag{11}$$

where $\mathscr{T} : \mathcal{U}(D; \mathbb{R}^{d_u}) \to \mathcal{V}(D; \mathbb{R}^{d_v})$ is a neural operator (or neural layer in the context of neural network) that shifts the channel dimension, transforming the original state $u^t$ into the latent state $v^t$, $\mathscr{P}_m : \mathcal{V}(D; \mathbb{R}^{d_v}) \to \mathcal{C}(D; \mathbb{R}^{d_m})$ is a neural operator that projects state onto a low-dimensional space, transforming the latent state $v^t$ into the reduced latent state $c^t$. $\mathscr{T}^{-1} : \mathcal{V}(D; \mathbb{R}^{d_v}) \to \mathcal{U}(D; \mathbb{R}^{d_u})$ and $\mathscr{P}_m^{-1} : \mathcal{C}(D; \mathbb{R}^{d_m}) \to \mathcal{V}(D; \mathbb{R}^{d_v})$ are neural operators that reverse the transformations of $\mathscr{T}$ and $\mathscr{P}_m$, respectively.

At the heart of the DyMixOp is the composite nonlinear operator $\mathscr{C} : \mathcal{C}(D; \mathbb{R}^{d_m}) \to \mathcal{C}(D; \mathbb{R}^{d_m})$. This comprises $L_d$ LGM layers that are integrated within a dynamics-informed network architecture, and are expressed as follows:

$$\mathscr{C}(c) = c_0 + \sum_{l=1}^{L_d} \Delta t_{c_{\theta_l}} \mathscr{F}_{\theta_l}(c_{l-1}), \tag{12}$$

where $\Delta t_{c_{\theta_l}}$ is a parameterized evolutionary step and the input $c_{l-1}$ to each layer is recursively defined as $c_{l-1} = \sigma(c_{l-2} + \mathscr{F}_{\theta_{l-1}}(c_{l-2})), l = 2, ..., L_d$, where $\sigma : \mathbb{R}^{d_m} \to \mathbb{R}^{d_m}$ is a nonlinear activation function applied element-wise, and the LGM layer $\mathscr{F}_{\theta_l}$ is defined as

$$\mathscr{F}_{\theta}(c) = \sum_{a=1}^{n_l} \mathscr{M}_{\theta}^{\mathcal{L}^a}(c) + \mathscr{H}_{\theta}\Big[ \sum_{b=1}^{n_n} \mathscr{M}_{\theta}^{\mathcal{N}^b}(c) \Big], \tag{13}$$

where $\mathscr{M}_{\theta}^{\mathcal{N}^b} : \mathcal{C}(D; \mathbb{R}^{d_m}) \to \mathcal{C}(D; \mathbb{R}^{d_m})$ and $\mathscr{M}_{\theta}^{\mathcal{L}^a} : \mathcal{C}(D; \mathbb{R}^{d_m}) \to \mathcal{C}(D; \mathbb{R}^{d_m})$ are nonlinear and linear LGM transformations, respectively, $n_n, n_l$ are the number of nonlinear and linear LGM transformations, respectively, and $\mathscr{H}_{\theta} : \mathcal{C}(D; \mathbb{R}^{d_m}) \to \mathcal{C}(D; \mathbb{R}^{d_m})$ is the local transformation. The nonlinear LGM transformation $\mathscr{M}_{\theta}^{\mathcal{N}}$ is defined as

$$\big(\mathscr{M}_{\theta}^{\mathcal{N}} c\big)(x) = (\mathscr{L}_{\theta}^{\text{loc}} c)(x) \odot (\mathscr{G}_{\theta}^{\text{glob}} c)(x), \tag{14}$$

where $(\mathscr{L}_{\theta}^{\text{loc}} c)(x) = \int_{P_\tau} p_\theta(x, y) c(y) \, dy$ and $(\mathscr{G}_{\theta}^{\text{glob}} c)(x) = \int_{E_\tau} e_\theta(x, y) c(y) \, dy$ are the local transformation and the global transformation, respectively, where $p_\theta : D \times P_\tau \to \mathbb{R}$ and $e_\theta : D \times E_\tau \to \mathbb{R}$ are kernel functions parameterized by $\theta$, and the integrals are well-defined over the domain $P_\tau$ and $E_\tau$, $\odot$ denotes the element-wise (Hadamard) product. The definition of linear operator $\mathscr{M}_{\theta}^{\mathcal{L}}$ is similar to the nonlinear one, but the global transformation or local transformation is specified as an indicator function $\mathbf{1}_D(x)$ depending on cases. Consequently, this formulation ensures that the DyMixOp captures both linear and nonlinear dynamics through local and global dependencies.

**Detailed Implementation for DyMixOp.** This work adopts key empirical insights from ML: local convolutions often outperform global spectral transforms, and applying activation functions to outputs typically boosts network performance. Accordingly, the global kernel $e_\theta$ in $\mathscr{M}^{\mathcal{L}^a}$ is fixed as $\frac{1}{E_\tau c}$, effectively reducing it to an indicator function $\mathbf{1}_D(x)$. Activations are applied at each intermediate state $c_{l-1}$. In the inertial manifold framework, $\mathscr{T}$ ideally maps to infinite dimensions but is practically set to twice the dimension of $\mathscr{P}_m$, whose optimal size remains a tunable hidden hyperparameter. For efficiency, $\mathscr{T}$, $\mathscr{P}_m$, and their inverses are implemented as local transforms—though this is flexible per use case. All transformations—$\mathscr{T}$, $\mathscr{P}_m$, their inverses, $\mathscr{H}_\theta$, and local components of $\mathscr{M}_{\theta}^{\mathcal{L}^a}$ and $\mathscr{M}_{\theta}^{\mathcal{N}^b}$—use 1×1 convolutions to ensure mesh-invariance. Global components in $\mathscr{M}_{\theta}^{\mathcal{N}^b}$ employ trainable truncated Fourier transforms, following FNO. For performance-critical tasks, larger convolutional kernels may be used—1×1 is not mandatory.

**Operator Approximation**. Computationally, the operator $\mathcal{G}$ must be discretized on the physical space and time. Given a discrete time sequence $\{t_i\}_{i=0}^{T}$ and states $u^{t_i}, u^{t_{i+1}} \in \mathcal{U}(D; \mathbb{R}^{d_u})$, sampled from a probability distribution $P_u$ over $\mathcal{U}(D; \mathbb{R}^{d_u})$, their relationship is defined as $u^{t_{i+1}} = \mathcal{G}(u^{t_i})$

for $i = 0, 1, \ldots, T - 1$. The objective is to approximate the operator $\mathcal{G}$ using the DyMixOp model parameterized by $\theta \in \Theta$, where $\Theta$ denotes the parameter space with dimensionality determined by the chosen architecture. Thus, the approximation of $\mathcal{G}$ using the DyMixOp is formulated as the following optimization problem:

$$\min_{\theta \in \Theta} \mathbb{E}_{(u^{t_i}, u^{t_{i+1}}) \sim P_u} ||\alpha[\mathcal{G}^{\dagger}(u^{t_i}; \theta) - u^{t_{i+1}}] + \beta[\mathscr{T}^{-1} \circ \mathscr{P}_m^{-1} \circ \mathscr{P}_m \circ \mathscr{T}(u^{t_i}) - u^{t_i}]||_{\mathcal{U}}, \quad (15)$$

where $\mathcal{G}^{\dagger}(u^{t_i}; \theta)$ represents the DyMixOp's prediction for $u^{t_{i+1}}$, $\|\cdot\|_{\mathcal{U}}$ denotes the norm in the Banach space $\mathcal{U}(D; \mathbb{R}^{d_u})$, and $\alpha, \beta$ are the weighted coefficients. The optimization target consists of the reconstruction and consistency errors. Although the DyMixOp architecture is designed based on the viewpoint of the evolution of dynamics, it presents a powerful capability to approximate the general solution operator mapping parameters to solutions. This will be demonstrated in the following experiments. In this work, we leverage data pairs of the form $\{[u^{t_{i-k}}, \ldots, u^{t_i}], u^{t_{i+1}}\}$ to account for temporal dependencies. The optimization problem is solved using empirical risk minimization, approximating the expectation with a finite dataset to effectively train the neural evolutionary operator.

## 4 RESULTS

### 4.1 EXPERIMENT SETTINGS

**Datasets**. We conduct experiments on the following datasets across multiple domains and PDE types: (i) 1D Kuramoto-Sivashinsky (KS), a one-dimensional parabolic PDE; (ii) 2D Burgers, a two-dimensional parabolic PDE; (iii) 2D CE-CRP, a two-dimensional hyperbolic PDE; (iv) 2D Darcy (Li et al., 2020a), a two-dimensional elliptic PDE; (v) 2D Navier-Stokes (NS), a two-dimensional parabolic PDE; (vi) 3D Brusselator (Bru.), a three-dimensional parabolic PDE; (vii) 3D Shallow Water (SW) (Cao et al., 2024), a three-dimensional hyperbolic PDE; More details on datasets can be found in the Appendix.

**Baselines**. We compare our method with several well-known models equipped with different transformations: (i) DeepONet (Lu et al., 2021a), a local-transformation-based architecture; (ii) GNOT (Hao et al., 2023), a global-transformation-based architecture; (iii) FNO (Li et al., 2020a), a LGA-transformation-based architecture; (iv) PeRCNN (Rao et al., 2023), a concise LLM-transformation-based architecture; (v) ConvLSTM (Shi et al., 2015), a precise LLM-transformation-based architecture. More details on baselines can be found in the Appendix.

**Experiment Details**. For the hyperparameters of the baselines and our methods, we assign different configurations for all models. By thoroughly exploring a wide range of configurations, we can comprehensively assess the models' potential. For fair comparisons, all models are trained for 500 epochs using the learning rate of $1e - 3$ and the AdamW optimizer (Loshchilov & Hutter, 2017) with 0.97 gamma and 6 step size. StepLR scheduler is utilized to modify the learning rate and a batch size 128 is used in the training on a single NVIDIA A100 GPU. More details about model configurations and implementation refer to Appendices

**Metrics**. All datasets are normalized in min-max normalization. For all datasets, except the 2D Darcy dataset, we adopt the mean squared error (MSE) as the reconstruction metric for training and evaluation. The 2D Darcy dataset employs the relative MSE instead, aiming to address extremely minimal solutions and prevent gradient vanishing.

### 4.2 MAIN COMPARISON RESULTS

The main experimental results for all datasets and methods are shown in Table 1. Based on these results, we have the following observations.

**An Excellent Performance and Generalization on Various PDE Types in the DyMixOp**. Our DyMixOp model consists of convection-inspired transformations and naturally embeds convection features into the model, enabling it to effectively capture convection dynamics with satisfactory performance. This perspective is validated by great gains in the 2D NS and 3D SW datasets, where these dynamics involve the influence of the convection term. The reduction in prediction error exceeds $75\%$ in both cases. It can also be noted in Fig. 2 that the prediction accuracy of the

Table 1: For each model, the metric on each dataset is the best one among almost **16** model configurations. For each dataset, the optimal result among the models is in bold, and the suboptimal result is underlined.

| Model | 1D KS | 2D Burgers | 2D CE-CRP | 2D Darcy | 2D NS | 3D Bru. | 3D SW |
|---|---|---|---|---|---|---|---|
| ConvLSTM | 0.8235 | 0.0326 | 0.0578 | 1.9e-6 | 0.0023 | 0.4744 | 7.2e-4 |
| PeRCNN | 1.0954 | 0.0520 | 0.0947 | 1.8e-5 | 0.0311 | 1.5628 | 1.5e-3 |
| GNOT | 1.7231 | 0.0316 | 0.0637 | 1.7e-7 | 0.0109 | 0.2042 | 6.8e-4 |
| DeepONet | 1.7337 | 0.0402 | 0.0629 | 1.7e-6 | 0.0120 | 2.6432 | 1.9e-3 |
| FNO | 0.0204 | 0.0020 | 0.0239 | 5.2e-9 | 0.0013 | 0.0599 | 8.3e-5 |
| DyMixOp | **0.0139** | **0.0007** | **0.0185** | **3.7e-9** | **0.0003** | **0.0538** | **1.1e-5** |
| Gain | 31.9% | 65.0% | 22.6% | 28.8% | 76.9% | 10.2% | 86.7% |

DyMixOp significantly improves as the model size increases. In contrast, other models show only slight performance improvements with larger sizes. The DeepONet model may even perform worse with a larger model size compared to a smaller one. Despite being inherently convection-inspired, the DyMixOp still presents a good generalization to other diffusion-dominated datasets such as the 1D KS and the 2D Darcy dataset, achieving an improvement beyond 25%. Similarly, the DyMixOp can reach better performances on these datasets as the model size increases.

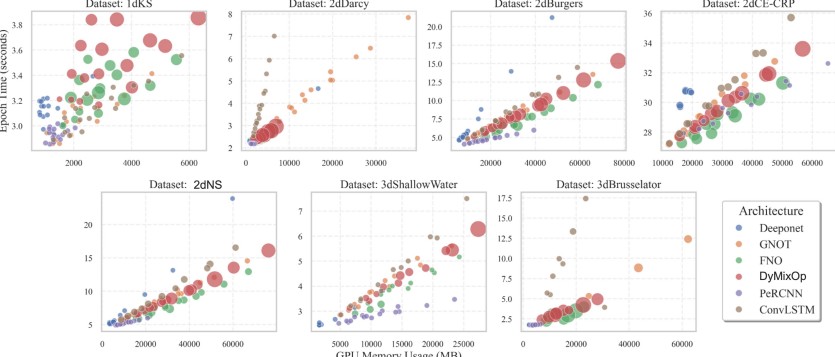

Figure 2: The performance of all model configurations across all datasets is visualized in a plane defined by training time per epoch and GPU memory usage. The size of each point indicates model performance, quantified as 1000 divided by the test dataset loss.

**Modest Training Time and Efficient GPU Memory Utilization in the DyMixOp.** Fig. 3 shows the GPU memory usage and training time of one epoch for all configurations of each model. For the time consumption, our DyMixOp generally consumes modest training time compared to other models of similar size, except for the 1D KS dataset. Due to the simplicity of the 1D case, the differences in time consumption among models are not particularly significant. For the GPU memory usage, the DeepONet and the PeRCNN models are restricted to their inherent architectures that are difficult to effectively extend to large model sizes, such that they usually require the minimal GPU memory usage. Instead, other models can readily cover kinds of model sizes. When the prediction errors (indicated by point size) are comparable, the DyMixOp typically exhibits similar time consumption and GPU memory usage as the DeepONet and PeRCNN models, and consumes less of both resources compared to the FNO, GNOT and ConvLSTM models. As the model size continues to increase, the DyMixOp model outperforms all models. These observations suggest that the DyMixOp model offers a compelling combination of high predictive accuracy, computational efficiency, and scalability, positioning it as a promising candidate for PDE solving.

### 4.3 ABLATION AND SCALING EXPERIMENTS

In this section, we conduct ablation and scaling experiments on the 2D NS dataset.

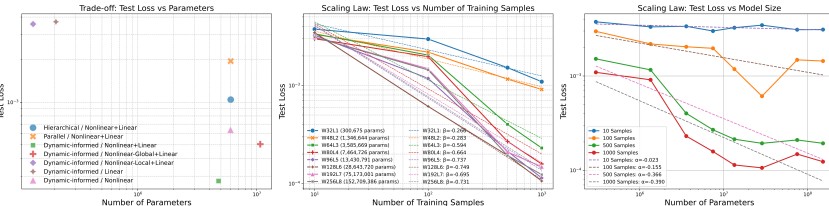

Figure 3: The illustration of scaling experiments for training samples and model sizes and ablation experiments. 'W' is the channel width and 'L' is the number of layers.

**Ablation Experiments**. We conduct ablation studies (Fig. 3) on nonlinear LGM transformations, LGM layers, and the dynamics-informed architecture, using standardized settings (e.g., 96-width channels unless noted). First, replacing the DyMixOp's nonlinear LGM transformations with purely local or global counterparts degraded performance: local-only failed to capture long-range dependencies, while global-only underperformed despite improvement, confirming that combining both is essential for modeling complex dynamics. Second, isolating linear and nonlinear components in LGM layers showed that linear-only performed poorly due to limited expressiveness, similar to local-only. Nonlinear-only achieved better results but collapsed around 340 epochs from instability, highlighting the need for a balanced integration of both transformation types. Lastly, testing the architecture design revealed that hierarchical stacking caused error accumulation and instability, while parallel stacking was stable but suboptimal. The combined parallel-hierarchical structure, guided by dynamical principles, delivered superior performance. These results emphasize the importance of integrating local and global features, balancing linear and nonlinear transformations, and adopting a hybrid architecture for effective modeling of complex dynamical systems.

**Scaling Experiments**. We adapt the task to non-autoregressive prediction to test a range of model sizes on a single NVIDIA A100 GPU. Eight configurations were examined, from W32L1 (0.3M parameters) to W256L8 (152.7M parameters), across data amounts of 10, 100, 500, and 1000 samples. Performance improved with more data across all model sizes, with larger models (e.g., W128L6) needing substantial data to optimize learning. Performance gains saturated around W128L6, particularly under data-rich regimes, highlighting the benefits of scaling. Spanning three orders of magnitude in size and data, these trends suggest further scaling could yield greater gains, affirming DyMixOp's potential for complex dynamical systems.

## 5 CONCLUSION

In this work, we propose **DyMixOp**, a principled neural operator framework that draws inspiration from complex dynamical systems to guide the architecture design for solving PDEs. Unlike prior approaches that often rely on heuristic or architecture-driven construction, our method is grounded in dynamical system theory, offering both theoretical rigor and empirical strength. The main contributions are summarized as follows: **(i) Theory-guided operator modeling**: We leverage **inertial manifold theory** to reduce infinite-dimensional nonlinear PDE dynamics into a finite-dimensional latent space, preserving essential nonlinear interactions. This dimensionality reduction offers a structured foundation for neural operator design with better physical interpretability and efficiency. **(ii) Convection-inspired Local-Global-Mixing (LGM) transformation**: Motivated by the local-global structure of convection terms in turbulence, we introduce a novel transformation that explicitly captures both local features and global interactions through element-wise mixing of local and global kernels. This mitigates spectral bias and enhances the model's expressiveness across scales. **(iii) Dynamics-informed architecture**: We construct a multi-layer architecture that mirrors the temporal evolution of dynamical systems in a hybrid variant. **(iv) Unified framework for complex PDEs**: By integrating these insights, DyMixOp delivers state-of-the-art performance across a range of PDE benchmarks, including convection-, diffusion-, and mixed-type equations.

This work demonstrates that **embedding physical and dynamical priors into neural operator design** is not only feasible but also impactful, pointing toward a more systematic path for developing generalizable and efficient neural solvers for complex PDE systems.

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
