# A Technical Appendices and Supplementary Material

## A.1 Datasets

To demonstrate the scalability and versatility of our method, we conduct experiments on the following datasets across multiple domains and PDE types. The input and output shapes in these datasets are conclude in Table 1.

Table 1: The illustration of the input shape, output shape in each case, where $c$ is the channel number representing the dimension of the vector, $t$ is the length of the temporal sequence and $s$ is the resolution size. The dataset sizes used for the training and test stage and the dataset type that attempt to approximate by neural operators are also shown.

| Case | Input Shape | Output Shape | Dataset Split | Dataset type |
|---|---|---|---|---|
| 1D KS | (c=1;t=10;s=256) | (c=1;t=10;s=256) | (1000;200) | Solutions $\rightarrow$ Solutions |
| 2D Burgers | (c=2;t=10;s=64$\times$64) | (c=2;t=10;s=64$\times$64) | (1000;200) | Solutions $\rightarrow$ Solutions |
| 2D CE-CRP | (c=5;t=10;s=64$\times$64) | (c=5;t=10;s=64$\times$64) | (1000;200) | Solutions $\rightarrow$ Solutions |
| 2D Darcy | (c=1;t=1;s=49$\times$49) | (c=1;t=1;s=49$\times$49) | (1000;200) | Parameters $\rightarrow$ Solutions |
| 2D NS | (c=1;t=10;s=64$\times$64) | (c=1;t=10;s=64$\times$64) | (1000;200) | Solutions $\rightarrow$ Solutions |
| 3D Brusselator | (c=1;t=1;s=39$\times$28$\times$28) | (c=1;t=1;s=39$\times$28$\times$28) | (800;200) | Parameters $\rightarrow$ Solutions |
| 3D SW | (c=2;t=10;s=64$\times$32) | (c=2;t=10;s=64$\times$32) | (1000;200) | Solutions $\rightarrow$ Solutions |

### A.1.1 1D Kuramoto-Sivashinsky

The Kuramoto-Sivashinsky equation is a nonlinear, fourth-order PDE, typically written as

$$u_t + uu_x + u_{xx} + u_{xxxx} = 0, \tag{1}$$

where $u$ is a scalar field representing a physical quantity, such as the height of an interface or a velocity perturbation, evolving over space $x$ (a 1D coordinate) and time $t$. $u_t$ is the time derivative and $u_x$ the space derivative. The presence of the second- and fourth-order spatial derivatives, named the diffusion term and hyperdiffusion term, gives it a dissipative, diffusion-like character, classifying it as a parabolic PDE, despite its nonlinear and chaotic behavior.

To generate the numerical solution as the dataset, the KS equation is solved by the pseudospectral method combined with the fourth-order exponential time-differencing Runge–Kutta formula, generating an 20-step temporal sequence $\{v_i\}_{i=1}^{20}$. The initial condition $v_0$ is sampled from $U(-1, 1)$. With $L = 64\pi$, the time integration is implemented starting from $250s$ when chaos is fully developed until the final time $T = 121$ where 2048 Fourier modes are employed to discretize the spatial domain. A total of $N = 5000$ temporal sequences are generated. Consequently, this dataset is downsampled to the resolution 256 and involves predicting how the $u$ evolves in later 10 steps from given initial 10 steps.

### A.1.2 2D Darcy

The 2d Darcy dataset models steady-state flow through porous media, such as groundwater movement or oil reservoir dynamics, using a linear elliptic PDE with spatially varying coefficients. The governing equation is:

$$-\nabla \cdot (k(x,y)\nabla u) = f(x,y), \tag{2}$$

where $u(x,y)$ represents the pressure or potential field across a 2D spatial domain defined by coordinates $x$ and $y$, $k(x,y) > 0$ is the permeability or diffusion coefficient that varies with position, and $f(x,y)$ is a source or sink term driving the flow (e.g., injection or extraction rates). The divergence $\nabla\cdot$ measures the net flow of the vector field $k\nabla u$. Physically, this equation balances the flux of $u$ scaled by $k$ with the source $f$, describing a steady-state system without time dependence. It is classified as an elliptic PDE.

The 2D Darcy equation is solved by using a second-order finite difference scheme on a $421 \times 421$ grid. The diffusion coefficient $k$ are sampled from $\psi\mathcal{N}\left(0, (-\Delta + 9I)^{-2}\right)$ with zero Neumann boundary conditions on the Laplacian where the mapping $\psi$ takes the value 12 on the positive part of the real line and 3 on the negative and the push-forward is defined pointwise. The forcing term $f$ is set to 1. This dataset is downsampled to the resoultion 49 and involves predicting $u(x,y)$ given $k(x,y)$.

### A.1.3   2D Burgers

The 2D Burgers dataset extends the classic 1D Burgers' equation to two dimensions, modeling viscous fluid flow with a nonlinear, time-dependent PDE. For a velocity vector $\mathbf{u} = (u, v)$, the system is:

$$\mathbf{u}_t + (\mathbf{u} \cdot \nabla)\mathbf{u} = \nu \Delta \mathbf{u}, \tag{3}$$

where $\mathbf{u} = (u, v)$ is the velocity field in the $x$ and $y$ directions, $\nu > 0$ is the viscosity coefficient. The nonlinear term $(\mathbf{u} \cdot \nabla)\mathbf{u}$ represents advection and $\nu \Delta \mathbf{u}$ is the diffusion term that smooths out sharp gradients due to viscosity. This equation is classified as a parabolic PDE.

To generate the numerical solution for the dataset, the 2D Burgers equation is solved numerically with a viscosity parameter $\nu = 0.005$. The spatial discretization employs a pseudospectral method on a $64 \times 64$ grid over the domain $[0, 1] \times [0, 1]$ with periodic boundary conditions. The spatial derivatives are computed using Fourier transforms, leveraging the efficiency of the pseudospectral approach. Time integration is performed using a fourth-order Runge-Kutta method with a time step of $\Delta t = 0.0025$, advancing the solution from $t = 0$ to $t = 0.5$ over 200 time steps. The solution is recorded at intervals of 0.025 time units, resulting in a temporal sequence of 21 time points (from $t = 0$ to $t = 0.5$, inclusive of the initial condition). The initial condition for each simulation is generated as a random field using a Fourier series with frequencies ranging from $-4$ to $4$ in each spatial direction. This field is normalized such that the maximum velocity magnitude across all batches and spatial points is scaled to approximately 1.5, and then shifted by a random constant vector with components uniformly distributed between $-1$ and $1$. Consequently, this dataset involves predicting how the $\mathbf{u}$ evolves in later 10 steps from given initial 10 steps.

### A.1.4   2D CE-CRP

The 2D CE-CRP dataset extends the stochastic four-quadrant Riemann problem to include curved subdomains, modeling inviscid fluid flow with the compressible Euler equations:

$$\frac{\partial}{\partial t} \begin{pmatrix} \rho \\ \rho\mathbf{u} \\ E \end{pmatrix} + \nabla \cdot \begin{pmatrix} \rho\mathbf{u} \\ \rho\mathbf{u} \otimes \mathbf{u} + p\mathbf{I} \\ (E + p)\mathbf{u} \end{pmatrix} = 0,$$

Therefore , it is classified as a hyperbolic PDEs. The domain is $[0, 1]^2$ with periodic boundary conditions, partitioned into four curved subdomains using random sine functions. Each subdomain has constant initial conditions for density $\rho$, velocity $\mathbf{u} = (u, v)$, and pressure $p$, sampled from uniform distributions: $\rho \sim U[0.1, 1]$, $u \sim U[-1, 1]$, $v \sim U[-1, 1]$, $p \sim U[0.1, 1]$. Unlike viscous models, the Euler equations permit discontinuities like shocks. The dataset contains 10,000 trajectories capturing the time evolution of the flow field and was simulated on the unit square up to $T = 1$. Details can refer to [4].

This dataset is saved in a $128 \times 128$ grid. In our work, it is downsampled to the resolution $64 \times 64$ and involves predicting how the 5-dimensional solution vector (i.e., density, horizontal velocity, vertical velocity, pressure, energy) evolves in later 10 steps from given initial 10 steps.

### A.1.5   2D Navier-Stokes

The 2D Navier-Stokes (NS) dataset models the dynamics of an incompressible, viscous fluid in two dimensions, governed by the Navier-Stokes equations. In this dataset, the equations are solved in the vorticity-stream function formulation, focusing on the evolution of the vorticity field $\omega = \nabla \times \mathbf{u}$, which in 2D satisfies:

$$\frac{\partial \omega}{\partial t} + (\mathbf{u} \cdot \nabla)\omega = \nu \Delta \omega. \tag{4}$$

The velocity field is recovered from the stream function $\psi$, where $\Delta \psi = -\omega$ and $\mathbf{u} = (\partial \psi / \partial y, -\partial \psi / \partial x)$. This equation is classified as a parabolic PDE.

To generate the numerical solution for the dataset, the 2D Navier-Stokes equation is solved using a pseudospectral method on a $256 \times 256$ grid over the periodic domain $[0, 2\pi) \times [0, 2\pi)$. Spatial derivatives are computed efficiently via Fourier transforms, and dealiasing is applied using the 2/3 rule to mitigate aliasing errors. Time integration is performed with the Crank-Nicholson method, a second-order implicit-explicit scheme, using a time step of $\Delta t = 0.001$. The simulation advances

from $t = 0$ to $t = 30$, with solutions recorded every 1 time unit from $t = 10$ to $t = 30$, resulting in a temporal sequence of 21 time points per trajectory.

Each simulation in the dataset begins with a random initial vorticity field $\omega_0$, where each grid point is independently sampled from a uniform distribution $U(-1, 1)$. The viscosity is set to $\nu = 10^{-5}$, promoting nearly inviscid behavior and the development of complex flow structures over time. The dataset comprises 1200 independent trajectories, each capturing the evolution of the vorticity $\omega$ across the 21 time points. Consequently, this dataset involves predicting how the voriticity $\omega$ evolves in later 10 steps from given initial 10 steps.

### A.1.6  3D Shallow Water

The 3D Shallow-water equations dataset models the dynamics of large-scale Rossby waves in the atmosphere, focusing on zonal flows in a viscous, rotating fluid layer over a spherical surface. For a fluid layer, the system is governed by the viscous Shallow-water equations:

$$\frac{\partial h}{\partial t} + \nabla \cdot (h\mathbf{V}) = 0, \tag{5}$$

$$\frac{\partial \mathbf{V}}{\partial t} + (\mathbf{V} \cdot \nabla)\mathbf{V} + f\mathbf{k} \times \mathbf{V} = -g\nabla h + \nu\Delta\mathbf{V} - k\mathbf{V}, \tag{6}$$

where $h$ is the fluid layer thickness, $\mathbf{V} = (u, v)$ is the velocity field in the eastward ($u$) and northward ($v$) directions, $f = 2\Omega \sin\phi$ is the Coriolis parameter, with $\Omega$ as the Earth's angular velocity, $g$ is the acceleration due to gravity, $\nu$ is the diffusion coefficient, $k$ is the viscous drag coefficient, $\mathbf{k}$ is the vertical unit vector. This system of hyperbolic partial differential equations (PDEs) is widely used to study atmospheric phenomena, such as Rossby waves, jet streams, and barotropic instabilities.

To generate the numerical solution for the dataset, the Shallow-water equations are solved using the Dedalus Project https://doi.org/10.3402/tellusa.v56i5.14436, a spectral method-based solver, for multiple cases with varying initial perturbation parameters, following the test case of a barotropically unstable mid-latitude jet as described in [1]. The simulation parameters are set as follows: Earth's angular velocity $\Omega = 7.292 \times 10^{-5}\,\text{s}^{-1}$, gravitational acceleration $g = 9.80616\,\text{m/s}^2$, hyperdiffusion coefficient $\nu = 1.0 \times 10^5\,\text{m}^2/\text{s}$ (matched at $\ell = 32$), maximum zonal velocity $u_{\text{max}} = 80\,\text{m/s}$, and jet boundaries $\phi_0 = \frac{\pi}{7}$, $\phi_1 = \frac{\pi}{2} - \phi_0$, with the jet's midpoint at latitude $\frac{\pi}{4}$. The spatial domain is a spherical grid spanning longitudes $\phi \in [0, 2\pi]$ and colatitudes $\theta \in [0, \pi]$, discretized on a $256 \times 128$ mesh, where 256 corresponds to the longitudinal direction and 128 to the latitudinal direction, reflecting the standard resolution for spherical coordinates where the colatitude range is half that of longitude. Time integration is performed from $t = 120$ to $t = 360$ hours with a time step of $\Delta t = 600$ seconds (equivalent to $\frac{1}{6}$ hours), and solutions are recorded every 12 hours, resulting in 20 time points per simulation.

For each case, the initial conditions consist of a zonal jet velocity profile defined over latitudes between $\phi_0$ and $\phi_1$, and a balanced height field computed via a linear boundary value problem (LBVP) solver to ensure geostrophic balance with the imposed jet. A localized Gaussian perturbation is then added to the height field to induce barotropic instability, parameterized by shape parameters $\alpha$ and $\beta$, which control the longitudinal and latitudinal extent of the perturbation, respectively. The perturbation is expressed as:

$$h'(\phi, \theta, t = 0) = h_{\text{pert}} \cos(\text{lat}) \exp\left[-\left(\frac{\phi}{\alpha}\right)^2\right] \exp\left[-\left(\frac{\text{lat} - \text{lat}_2}{\beta}\right)^2\right], \tag{7}$$

where latitude $\text{lat} = \frac{\pi}{2} - \theta$, the perturbation center is at $\text{lat}_2 = \frac{\pi}{4}$, the perturbation amplitude is $h_{\text{pert}} = 120\,\text{m}$, $\phi$ is the longitude, and $\theta$ is the colatitude. The parameters $\alpha$ and $\beta$ are systematically varied across a grid of values: $\alpha$ ranges from $\frac{1}{120}$ to 10 over 40 evenly spaced points, and $\beta$ ranges from $\frac{1}{300}$ to 2 over 30 evenly spaced points, yielding a total of $40 \times 30 = 1200$ unique simulations. These ranges allow the perturbation to vary from highly localized (small $\alpha$ and $\beta$) to broadly spread (large $\alpha$ and $\beta$) in both longitudinal and latitudinal directions. The dataset comprises 1200 simulations, and the spatial fields, originally on a $256 \times 128$ grid, are downsampled to the resolution $64 \times 32$. Each simulation records snapshots of the height field $h$ and vorticity $\omega$ every 12 hours, saving up to 30 snapshots per case. Consequently, this dataset involves predicting how the height and velocity fields evolve in later 10 steps from given initial 10 steps.

 **A.1.7  3D Brusselator**

 The 3D Brusselator dataset models autocatalytic chemical reactions, capturing the spatiotemporal
evolution of reactant concentrations through a system of nonlinear reaction-diffusion equations. For
concentration fields $u$ and $v$, the system is defined as:

$$\frac{\partial u}{\partial t} = D_0\Delta u + a + f(t) - (b+1)u + u^2v, \tag{8}$$

$$\frac{\partial v}{\partial t} = D_1\Delta v + bu - u^2v, \tag{9}$$

where $u$ and $v$ represent the concentrations of two chemical species at location $\mathbf{x} = (x, y)$ and time $t$,
$D_0$ and $D_1$ are the diffusion coefficients for $u$ and $v$, respectively, $a$ and $b$ are constant parameters
representing fixed concentrations, $f(t)$ is a time-dependent signal introducing fluctuations to the
system, and the nonlinear term $u^2v$ accounts for the autocatalytic reactions. This system is classified
as a parabolic PDEs.

To generate the numerical solution for the dataset, the 3D Brusselator reaction-diffusion equations
are solved numerically with the following parameters: $a = 1$, $b = 3$, $D_0 = 1$, and $D_1 = 0.5$.
The spatial domain is a periodic square $[0, 1) \times [0, 1)$, discretized on a $28 \times 28$ grid. The time
evolution is computed using the finite difference method implemented in the py-pde solver [14], with
a computational time step of $\Delta t = 0.02$ seconds, spanning the time range from $t = 0$ to $t = 20$.
Snapshots of the solution are recorded at intervals of $0.5$ seconds, resulting in a temporal sequence of
38 time points per trajectory. Each simulation produces a spatiotemporal dataset of size $28 \times 28 \times 38$
for the concentration fields.

The initial conditions are defined as $u(\mathbf{x}, 0) = 1$ and $v(\mathbf{x}, 0) = 1 + \epsilon(\mathbf{x})$, where $\epsilon(\mathbf{x})$ is a randomly
generated spatially varying field across the $28 \times 28$ grid. A time-dependent signal $f(t)$ introduces
fluctuations and differs between the training and testing datasets. For the training set, the signal
is $f_{\text{train}}(t) = A_{\text{train}}e^{-0.01t}\sin(t)$, while for the testing set, it is $f_{\text{test}}(t) = A_{\text{test}}e^{-0.05t}\sin(t)$. The
amplitude $A$ is randomly sampled from the interval $[0.01, 10]$, with 800 distinct values used for
$A_{\text{train}}$ in the training set and 200 values for $A_{\text{test}}$ in the testing set. This dataset is designed to predict
the evolved concentration field $u(\mathbf{x}, t)$ in three dimensions (2D space + 1D time) given the signal
$f(t)$. To adapt to the network architecture, the one-dimensional signal $f(t)$ is extended to a three-
dimensional field $f(\mathbf{x}, t)$, where the value of $f$ remains constant across the spatial dimensions for
each corresponding time point.

**A.2  Baselines**

In the following models, they only take the previous temporal solutions and spatial coordinates as the
generic input for a fair comparison. Additional prior informations applied in some models, such as
POD-modulated inputs [2] and edges [11], are eliminated. All models predict temporal solutions for
future multiple steps in an auto-regressive way unless specially specified. The applied convolutional
transformations are kept with the kernel size of 1, unless specially specified, for retaining the mesh
independency which is the important property of the neural operator. In such a strict restriction, the
investigation on the potential of the model architecture is more reliable.

**A.2.1  DeepONet**

The DeepONet is a high-level defined model composited of the branch network [9], which takes the
previous temporal solutions as input, and the trunck network, which takes the spatial coordinates
corresponding to their solutions in branch network as input. In the DeepONet architecture, it must
take the large kernel size to subsample the solution resolution. Although it can be generalized to
the predictions in other resolutions, it depends on the non-align training, incorporating the targeted
resolution data different from the input resolution in the training. From the practical training
viewpoint, it slightly deviates from the scope of the neural operator, where usually take the resolution
in the input for the training and then can generalized to other resolutions. The branch network is
built by stacking $n$ convolutional layers with the kernel size of 3 following a flatten layer, two fully
connected neural layers (FCNL) connected with a GELU activation function. When the approximated
solutions are n-dimensional vector (e.g., the height and vorticity in the 3D Shallow Water case),
the number of the branch networks is increased to $d_n$ for outputting different component. These

components finally work with the output from the trunk network to form a $d_u$-dimensional solution at next step. The trunk network is built by stacking $n$ FCNLs, achieving the targeted width channel in a quadratic growth. The point-wise FCNLs and convolutional layers suggest the DeepONet is the local-transformation-based architecture. It is worth to note that the GPU memory utilization becomes considerably large when the number of layers, $n$, is small. The shallow architecture is not only difficult to effectively extract high-level features but also remains pretty large resolutions. It finally leads to a high GPU utilization but a worse performance. For a superior performance, one need to carefully design the architecture, causing a difficulty in a quick usability. The illustration of the DeepONet architecture is shown in Table 2.

Table 2: The illustration of the DeepONet[10]. $Flatten$ means the flatten layer which flattens the multidimensional tensor into an one-dimensional tensor with $d_{fs} = s \times s \times d_c$ where $s$ is the resolution size operated by all convolutional layers. $d_{space}$ means the spatial dimension in the case. $FCNL$ means the fully connected neural layer, $Conv$ means the convolutional layer, $k$ the kernel size, $p$ the padding size, $s$ the stride size, $BN$ means the batch normalization layer and GELU means the GELU activation function.

| Layer | Branch Network(s) | Trunk Network |
|---|---|---|
| 1 | $[Conv(d_i \rightarrow d_c, k = 3, p = 1, s = 2), BN, \text{GELU}]$ | $[FCNL(d_{space} \rightarrow d_c//(2^{n-1})), \text{GELU}]$ |
| ... | ... | ... |
| n | $[Conv(d_c \rightarrow d_c, k = 3, p = 1, s = 2), BN, \text{GELU},$ $Flatten, FCNL(d_{fs} \rightarrow d_{fs}//2^{d_{space}}), \text{GELU},$ $FCNL(d_{fs}//2^{d_{space}} \rightarrow d_c)]$ | $[FCNL(d_c//2 \rightarrow d_c), \text{GELU}]$ |

### A.2.2 ConvLSTM

The ConvLSTM is a convolution-modified version for the classical LSTM architecture [13], firstly proposed by [12]. The input for each ConvLSTM cell is the spatial coordinates, and the initial hidden state and initial cell state are repeatedly filled with the previous temporal solutions until reaching the hidden dimension $d_c$. The activation functions is remained for keeping the original intent in the LSTM, expressing an effective combination of the long term and short term memory effort. The number of layers indicates the number of the sequence of ConvLSTM cells. The length of the sequence of ConvLSTM cells is the prediction steps. The ConvLSTM cell merges the current input and hidden state through concatenation along the channel dimension, then applies a convolutional layer to produce the input, forget, and output gates, as well as the cell input. These elements are transformed using sigmoid activations for the gates and a tanh activation for the cell input, enabling the cell to selectively update its state. The cell state is adjusted by balancing retained information and new input, while the next hidden state is derived by modulating the transformed cell state with the output gate and considered the solution for one future step. The element-wise products in the balance for the history and new information and the modulate for the output suggest the ConvLSTM is the LLM-transformation-based architecture.

### A.2.3 FNO

The FNO [7] is a neural operator composed of the lifting layer, projection layer and the Fourier neural layers, where Fourier neural layer is the addition of the convolutional transformation and the parameterized Fourier transformation. The FNO starts with the initial lifting layer that transforms the generic input into a higher-dimensional space. This is followed by multiple layers, each integrating a spectral global transformation in Fourier space and a local convolutional layer. The spectral global transformation selectively processes specific frequency bands, applying complex multiplications to these bands and truncating high-frequency components. The outputs of the spectral global transformation and local convolutions are combined, followed by batch normalization (except in the final layer) and GELU activation. Finally, the processed data is projected back to the target output dimension using the projection layer. The number of layers indicates the number of Fourier neural layers. The addition of the spectral global transformation and the local convolutional transformtion suggest the FNO is the LGA-transformation-based architecture. The applied number of truncated modes in all cases are listed in Table 3.

Table 3: The applied number of the truncated modes in each case.

| Case | Truncated modes |
|------|:---------------:|
| 1D KS | 32 |
| 2D Burgers | 12 |
| 2D CE-CRP | 12 |
| 2D Darcy | 12 |
| 2D NS | 12 |
| 3D Brusselator | 4 |
| 3D SW | 12 |

### A.2.4 GNOT

The Generalized Neural Operator Transformer (GNOT) architecture is a transformer-based neural network designed for solving partial differential equations and operator learning tasks [2]. It originally processes spatial coordinates and multiple input functions through separate multi-layer perceptrons to create embeddings. In this work, it only involve the spatial coordinates and previous temporal solutions. These embeddings are then refined through a series of cross-attention blocks, where cross-attention integrates information from the input into the spatial features, followed by self-attention to capture spatial dependencies. A mixture of experts mechanism adaptively applies transformations based on spatial coordinates, enhancing flexibility across different domain regions. Finally, an output multi-layer perceptron generates the solution field. The transformer-based architecture suggests the GNOT is the global-transformation-based architecture. The number of layers indicates the number of the block that combines the cross-attention and mixture experts mechanism. Notably, GNOT architecture outputs all future temporal solutions once due to the requirement for a large GPU memory. Parameters in Table are applied in all cases.

Table 4: The applied parameters in all cases. $n$ means the specified number of the block that combines the cross-attention and mixture experts mechanism.

| Param. | value |
|--------|:-----:|
| Number of attention head | 1 |
| Number of experts | 2 |
| Inner dimension of experts | 4 |
| Number of MLP | n |

### A.2.5 PeRCNN

The GNOT architecture is a recurrent convolutional neural network designed for solving time-dependent partial differential equations, integrating convolutional layers to capture spatial dynamics and recurrent mechanisms to model temporal evolution. It employs parallel convolutional layers to process input fields, followed by pointwise convolutions to produce outputs, enabling flexible handling of multi-channel spatial data. The network incorporates finite difference-based derivative operators, including Laplacian and gradient computations, to enforce physical constraints, enhancing its ability to learn complex spatiotemporal patterns efficiently. In this work, the finite difference-based derivative operators is be eliminated to keep the mesh independency property. The number of layers indicates the number of the parallel convolutional layers. The product operator by pointwise convolutions suggests the PeRCNN is the LLM-transformation-based architecture. Parameters applied in each case are listed in Table.

### A.3 Experiment Details

For the hyperparameters of the baselines and our methods, we specify the network width from $\{32, 48, 64, 96\}$ and the number of layers from $\{1, 2, 3, 4\}$, except for some configurations are beyond GPU memory. By traversing as possible as different configurations, we can comprehensively evaluate the potentials of models. All applied configurations are listed in Table 5. For fair comparisons, all

models are trained for $500$ epochs using the learing rate of $1e-3$, the GELU activation function [3] and the AdamW optimizer [8] with $0.97$ gamma and $6$ step size. StepLR scheduler are utilized to modify the learning rate and a batch size of $128$ is used in the training on one NVIDIA A100 GPU. Especially, achieving a satisfactory performance of the PeRCNN model usually requires a large number of epochs due to its considerably simple architecture. Therefore, more epochs are used to train the model such that the total consumption of the training time is close to other models. Details of epochs and scheduler for PeRCNN model can be found in Table 6.

## A.4 Metrics

Using the combination of the reconstruction and consistency loss function, it leads to a conflict effect between these two loss functions at the late stage of the training, resulting in a higher error than the pure reconstruction loss function. Therefore, in this work, the reconstruction loss function is applied alone, i.e., $\alpha = 1, \beta = 0$. A suitable coefficient setting still requires comprehensive investigation in the future. Alternatively, pretraining the consistency loss function on the dimension-shifting layer, projection layer and their inverse layers. Then freezing these parameters, performing the training on the LGM layers with the reconstruction loss function alone. The two-stage training strategy is helpful to promote the convergent speed in the traning of the LGM layers.

## A.5 Coefficient Dynamics in Spectral Space

Although the reduced latent dynamics is built, it still evolves in the $d_v$-dimensional space and is difficult to perform calculation in practice. By expanding the reduced latent state $\hat{v}$ onto the spectral space, it has

$$\hat{v} = \sum_{i=1}^{m} c_i w_i, \tag{10}$$

where $w_i \in \mathbb{R}^{d_v}$ is one spectral basis of the first $d_m$ eigenmodes and $c_i \in \mathbb{R}$ is its corresponding spectral coefficient. Substituting this expansion into Eq. **??** and using the linearity of $\mathcal{L}$, the orthonormality $\langle w_i, w_k \rangle = \delta_{ik}$ and the inner product with $w_k$ for $k = 1, \ldots, d_m$:

$$\frac{\partial c_k}{\partial t} = \sum_{i=1}^{d_m} c_i \langle \mathcal{L} w_i, w_k \rangle + \left\langle \mathcal{N} \left( \sum_{j=1}^{d_m} c_j w_j \right), \mathscr{P}_m w_k \right\rangle + \left\langle \mathcal{R} \left[ \mathcal{N} (\sum_{j=1}^{m} c_j w_j) \right], \mathscr{P}_m w_k \right\rangle \tag{11}$$

Since $w_k$ is in the range of $\mathscr{P}_m$, $\mathscr{P}_m w_k = w_k$. Finally, we can give the dynamics of coefficient $c \in \mathcal{C}(D; \mathbb{R}^{d_m})$, where $\mathcal{C}(D; \mathbb{R}^{d_m})$ is a Banach space, in a compact vector form:

$$\frac{\partial c(t)}{\partial t} = \mathcal{L}_c c + \mathcal{N}_c(c) + \mathcal{R}_c[\mathcal{N}(c)], \tag{12}$$

where $\mathcal{L}_c : \mathcal{C}(D; \mathbb{R}^{d_m}) \to \mathcal{C}(D; \mathbb{R}^{d_m})$ is a linear operator for $c$ with its $k$-th component $[\mathcal{L}_c]_k = \langle \mathcal{L} w_m, w_k \rangle$, $\mathcal{N}_c : \mathcal{C}(D; \mathbb{R}^{d_m}) \to \mathcal{C}(D; \mathbb{R}^{d_m})$ is a nonlinear operator for $c$ with its $k$-th component $[\mathcal{N}_c]_k = \left\langle \mathcal{N} \left( \sum_{j=1}^{m} c_j w_j \right), \mathscr{P}_m w_k \right\rangle$, and $\mathcal{R}_c : \mathcal{C}(D; \mathbb{R}^{d_m}) \to \mathcal{C}(D; \mathbb{R}^{d_m})$ is a nonlinear operator for $c$ with its $k$-th component $[\mathcal{R}_c]_k = \left\langle \mathcal{R}[\mathcal{N}(\sum_{j=1}^{m} c_j w_j)], \mathscr{P}_m w_k \right\rangle$. Assume $\mathcal{R}_c[\mathcal{N}_c(c)] \approx \mathcal{R}_c[\mathcal{N}(c)]$ and $\mathcal{A}[\mathcal{N}_c(c)] = \mathcal{N}_c(c) + \mathcal{R}_c[\mathcal{N}_c(c)]$, Eq. 12 becomes a more compact form:

$$\frac{\partial c(t)}{\partial t} \approx \mathcal{L}_c c + \mathcal{A}[\mathcal{N}_c(c)] = \mathscr{F}(c), \tag{13}$$

where $\mathcal{A} : \mathcal{C}(D; \mathbb{R}^{d_m}) \to \mathcal{C}(D; \mathbb{R}^{d_m})$ is an operator mapping the low-mode component to the full-mode component, $\mathscr{F} : \mathcal{C}(D; \mathbb{R}^{d_m}) \to \mathcal{C}(D; \mathbb{R}^{d_m})$ is a nonlinear operator that acts on $c$ and represents the dynamics of $c$.

## A.6 Dynamics-informed architecture

We need to assume that the transformation $\mathscr{T}$ induces a linear time transformation $t_v = g(t_u)$ and the projection $\mathscr{P}_m$ maintains the time scale $t_v = t_c$. It indicates a relationship between states

Table 5: The configurations of neural layers and layer widths for each model on each dataset. When the parameter does not come with the underline, then each layer parameter coupled with each width parameter to form a configuration for the model, such as total 16 configurations applied for the DyMixOp model on the 1D KS dataset. The underline of the layer and width represents the absence of this configuration, such as the configuration of layer 1 and width 96 is not utilized to the DeepONet model on the 2D Burgers dataset.

| Dataset | Network Param. | DyMixOp | FNO | ConvLSTM | PeRCNN | GNOT | DeepONet |
|---|---|---|---|---|---|---|---|
| 1D KS | Layer | 1/2/3/4 | 1/2/3/4 | 1/2/3/4 | 1/2/3/4 | 1/2/3/4 | 1/2/3/4 |
| | Width | 32/48/64/96 | 32/48/64/96 | 32/48/64/96 | 32/48/64/96 | 32/48/64/96 | 32/48/64/96 |
| 2D Burgers | Layer | 1/2/3/4 | 1/2/3/4 | 1/2/3/4 | 1/2/3/4 | 1/2/3/4 | 1/2/3/4 |
| | Width | 32/48/64/96 | 32/48/64/96 | 32/48/64/96 | 32/48/64/96 | 32/48/64/96 | 32/48/64/96 |
| 2D CE-CRP | Layer | 1/2/3/4 | 1/2/3/4 | 1/2/3/4 | 1/2/3/4 | 1/2/3/4 | 3/4 |
| | Width | 32/48/64 | 32/48/64 | 32/48/64 | 32/48/64 | 32/48/64 | 32/48/64 |
| 2D Darcy | Layer | 1/2/3/4 | 1/2/3/4 | 1/2/3/4 | 1/2/3/4 | 1/2/3/4 | 1/2/3/4 |
| | Width | 32/48/64/96 | 32/48/64/96 | 32/48/64/96 | 32/48/64/96 | 32/48/64/96 | 32/48/64/96 |
| 2D NS | Layer | 1/2/3/4 | 1/2/3/4 | 1/2/3/4 | 1/2/3/4 | 1/2/3/4 | 1/2/3/4 |
| | Width | 32/48/64/96 | 32/48/64/96 | 32/48/64/96 | 32/48/64/96 | 32/48/64/96 | 32/48/64/96 |
| 3D Brusselator | Layer | 1/2/3/4 | 1/2/3/4 | 1/2/3/4 | 1/2/3/4 | 1/2/3 | 3/4 |
| | Width | 16/32 | 16/32 | 16/32 | 16/32 | 16 | 16/32 |
| 3D Shallow Water | Layer | 1/2/3/4 | 1/2/3/4 | 1/2/3/4 | 1/2/3/4 | 1/2/3/4 | 1/2/3/4 |
| | Width | 32/48/64 | 32/48/64 | 32/48/64 | 32/48/64 | 32/48/64 | 32/48/64 |

Table 6: The number of training epochs and the step size of learning rate scheduler applied by the PeRCNN model on each dataset.

| Model | Dataset | Epochs | Step Size |
|---|---|---|---|
| PeRCNN | 1D KS | 1500 | 20 |
| | 2D Burgers | 2500 | 33 |
| | 2D CE-CRP | 500 | 6 |
| | 2D Darcy | 5500 | 73 |
| | 2D NS | 4000 | 53 |
| | 3D Brusselator | 10000 | 133 |
| | 3D SW | 2000 | 26 |

$\mathscr{P}_m \mathscr{T}(u)(t_u) = \mathscr{P}_m v(t_v) = c(t_c)$, and their dynamics:

$$\frac{\partial \mathscr{P}_m \mathscr{T}(u)(t_u)}{\partial t_u} = \frac{\partial \mathscr{P}_m v(g(t_u))}{\partial g(t_u)} \frac{\partial g(t_u)}{\partial t_u} = \zeta \frac{\partial c(t_c)}{\partial t_c} = \zeta \mathscr{F}, \tag{14}$$

where $\zeta = \frac{\partial g(t_u)}{\partial t_c}$ is the scaling factor, transforming the time scale $t_u$ to $t_c$. When dynamics are represented by neural layers, let we assume the scaling factor is the number of stacked LGM layers, $\zeta = L_d$, and Eq. 14 can be parameterized and expanded as

$$\zeta \mathscr{F}_\theta(c) = \mathscr{F}_{\theta_1}(c) + \cdots + \mathscr{F}_{\theta_{L_d}}(c), \tag{15}$$

where $\mathscr{F}_{\theta_l}, l = 1, ..., L_d$ is parameterized by neural layer $l$. Assuming an initial reduced latent state $c_0$ and an evolutionary step $\Delta t_c = \Delta t_u / L_d$ to the target reduced latent state $c_{L_d}$, the form of Eq. 15 suggest two alternatives to constitute the network architecture: a parallel way, stacking all layers which consider the same input and is formulated as

$$c_{L_d}(x) = c_0(x) + \Delta t_c \sum_{l=1}^{L_d} \mathscr{F}_{\theta_l}(c_0), \tag{16}$$

or a hierarchical way, connecting all layers in order, each of which takes the output from the previous one as the input and is formulated as

$$c_l(x) = c_{l-1}(x) + \Delta t_c \mathscr{F}_{\theta_l}(c_{l-1}), \quad l = 1, ..., L_d. \tag{17}$$

In the parallel way, it facilitates easier gradient propagation to the preceding neural layers, mitigating the issues of gradient vanishing or exploding while treating all parameterized dynamics with similar importance. In the hierarchical way, it builds finer dynamics by leveraging the previous reduced latent state $c_{l-1}$. This process progressively refines the model's accuracy, as each successive layer adjusts the representation established by the prior dynamics and finally form the successive trajectory from the initial state to the terminal state. Taking full advantages of two ways, a combined connection version is proposed.

In this version, we break the limitation that each reduced latent dynamics evolves with a fixed step $\Delta t_c$, and parameterize the evolutionary step $\Delta t_{c_{\theta_l}}$, providing the flexibility to represent the reduced latent dynamics of $c$. Consequently, we can get the following formula of the output:

$$c_{L_d} = c_0 + \sum_{l=1}^{L_d} \Delta t_{c_{\theta_l}} \mathscr{F}_{\theta_l}(c_{l-1}) \tag{18}$$

where the input to each layer $c_{l-1}$ is recursively defined as

$$c_{l-1} = c_{l-2} + \mathscr{F}_{\theta_{l-1}}(c_{l-2}), l = 2, ..., L_d. \tag{19}$$

Notably, $\{c_l\}_{l=1,...,L_d-1}$ are computed using a default evolutionary step $\Delta t_v = 1$. When it comes to the calculation of the final output $c_{L_d}$, the parameterized evolutionary step $\Delta t_{c_{\theta_l}}$ is incorporated.

## A.7 Ablation experiments

In this section, we conduct ablation experiments on the 2D NS dataset. These experiments focus on the impact of the nonlinear LGM transformations, the LGM layers and dynamics-informed architecture. The configuration of 4 layers and 72 widths is specified as the baseline.

**Impact of the nonlinear LGM transformations.** In this ablation study, we explore the significance of nonlinear LGM transformations within the DyMixOp. To isolate their impact, we replace the nonlinear LGM transformations with either purely local transformations or purely global transformations, while preserving the rest of the architecture. To ensure a fair comparison, all model configurations are standardized to 96 width channels, maintaining comparable GPU memory budgets across the variants. As depicted in Fig. 3 in the maintext, the scenario with purely local transformations exhibits the weakest performance. The absence of global information likely limits the model's ability to capture long-range dependencies essential for modeling complex nonlinear dynamics. The scenario with purely global transformations outperforms the local-only case but remains suboptimal. While global transformations effectively handle domain-wide relationships, they may overlook fine-grained, localized interactions critical for accurate nonlinear approximations. The LGM transformations, achieves the highest performance. This highlights the advantage of combining multi-scale features to approximate nonlinear dynamics effectively. These findings confirm that LGM transformations enhance the neural operator's capacity to model nonlinear dynamics by integrating localized and global perspectives in a multiplicative framework.

**Impact of the LGM layers.** The LGM layers comprise two key components: nonlinear LGM transformations, which approximate nonlinear dynamics, and linear LGM transformations, which approximate linear dynamics. To evaluate their individual contributions, we conducted an ablation study by removing one type of transformation at a time and adjusting the GPU memory usage to ensure a fair comparison. Specifically, for the case with only linear LGM transformations, we increased the GPU memory count by setting the width to 142. For the case with only nonlinear LGM transformations, we retained the original configuration. As illustrated in Fig. 3 in maintext, the model relying solely on linear LGM transformations performs poorly, resembling outcomes seen with only local transformations. This degraded performance likely stems from its inability to capture nonlinear dynamics. In contrast, the model using only nonlinear LGM transformations achieves better performance, supporting the above viewpoint. However, training with only nonlinear LGM transformations encountered issues, halting around 340 epochs. This interruption may occur because nonlinear transformations struggle to simultaneously represent both linear and nonlinear dynamics effectively. These results highlight the necessity of integrating both linear and nonlinear LGM transformations within the LGM layers. Combining these transformations ensures robust training and delivers superior performance, striking a balance that neither component can achieve alone.

**Impact of the dynamics-informed architecture.** The dynamics-informed architecture combines parallel and hierarchical layer stacking approaches to enhance model performance. To validate the effectiveness of this combined strategy, we conducted ablation experiments by isolating the parallel and hierarchical approaches and comparing them to the original configuration. As illustrated in Fig. 3 in maintext, the hierarchical approach exhibited unstable training with noticeable fluctuations, ultimately yielding the poorest performance among the tested configurations. The instability likely arises from error accumulation or poor gradient flow through the deep sequential stack. The parallel approach, by contrast, demonstrated stable training and outperformed the hierarchical approach. Its success can be attributed to the ability to capture diverse features without the risk of error propagation inherent in sequential processing. The dynamics-informed architecture, by simultaneously combining parallel and hierarchical layer stacking and embedding the prior knowledge of dynamics, provides a robust framework for modeling complex dynamical systems within the DyMixOp. Ablation experiments confirm that this hybrid approach outperforms its individual components, highlighting its potential for advancing deep learning applications in dynamical systems.

## A.8 Scaling experiments

In this section, we conduct scaling experiments on the 2D Navier-Stokes (NS) dataset to explore how model performance scales with data amount and model size. Here, we adapt the autoregressive task to a non-autoregressive one, necessitating models to produce all evolutionary predictions in a single pass, thereby encompassing a broader range of model sizes for a thorough examination on one NVIDIA A100 GPU. A total of 8 configurations were examined, encompassing W32L1 with 0.3M parameters, W48L2 with 1.4M parameters, W64L3 with 3.6M parameters, W80L4 channels with 7.5M parameters, W96L5 with 13.4M parameters, W128L6 with 28.6M parameters, W192L7 with 75.1M parameters, and W256L8 with 152.7M parameters, where WxLy denotes x width channels and y layers in brief.

**Scaling impact of the data amount.** We assess model performance across four data amounts: 10, 100, 500, and 1000 samples. As illustrated in Fig. 3 in maintext, performance improves across all configurations as the data amount increases. The scaling trends are visualized through fitted lines for each model size, revealing key insights: The slope of the fitted line is minimized at the W128L6 configuration, indicating that larger models require more substantial data amounts to effectively learn high-level representations.

**Scaling impact of the model size.** We further evaluate how performance scales with model size across the eight configurations. As shown in Fig. 3 in maintext, a similar conclusion that performance consistently improves with increasing model size until reaching W128L6 can be draw. The slope of the performance versus model size curve decreases as data amount grows, suggesting that abundant data amplifies the benefits of scaling up model size.

While these experiments are limited by computational resources—spanning approximately three orders of magnitude in both model size and data amount—the results indicate promising scaling capabilities for the DyMixOp model. The observed trends suggest that further increases in model size and data amount could drive additional performance improvements, underscoring the DyMixOp's potential for tackling complex dynamical systems.

## A.9    Limitation and discussion

**Universal Approximation Theorem** While our method is firmly grounded in inertial manifold theory, offering a principled reduction of infinite-dimensional PDE dynamics into a finite latent space, we acknowledge a theoretical gap and our proposed architecture does not yet come with a formal UAT guarantee. But we can still have a look at the realizability of this theorem. We embed a sufficiently expressive operator network FNO block inside our latent dynamics loop. Since such blocks are UAT-capable, the overall architecture would inherit universality in principle. This suggests that, by constructing our latent-update functions and LGM transforms such that they can emulate any continuous mapping on the latent space, the entire operator pipeline could satisfy an approximation theorem in the limit of sufficiently high capacity. Similar reasoning is often used in works like FNO and DeepONet papers to infer universality from the constituent blocks [5, 9]. While empirical performance across varied PDE benchmarks strongly supports its effectiveness, a formal universal approximation guarantee for the complete architecture remains to be established in future theoretical work.

**Irregular grid** In this implemented DyMixOp, the global transformation is the parameterized Fourier transformation, which require the uniform grid and may encounter the difficulty of a direct implementation in the irregular grid. However, there still exits some methods to solve this problem in the existing literature, for example, transforming the irregular grid in the physical space to the regular grid in the computational space [6]. Alternatively, the global transformation can be specified as the transformer architecture, then DyMixOp can naturally deal with the irregular grid problems.

**Weak performance in small model size** The results shown in Section Main comparison results in the maintext indicates that the performance is weaker than the FNO model when the model size is around one or two layers. It may be caused by the inadequate evolution in the reduced latent dynamics from the perspective of the complex dynamical system, or by the inadequate high-level feature abstract from the perspective of the machine learning. This feature is similar to the transformer architecture, and scaling experiments indicates the promising potential of the DyMixOp when it can be trained with the large model size and large train dataset.