# OpenReview forum: "DyMixOp: Guiding Neural Operator Design for PDEs from a Complex Dynamics Perspective with Local-Global-Mixing"
_ICLR.cc/2026/Conference — ICLR 2026 Conference Withdrawn Submission_

### Official Review · Reviewer_KBnD · 2025-10-19

**Soundness:** 2
**Presentation:** 2
**Contribution:** 2
**Rating:** 2
**Confidence:** 4

**Summary:**

This paper proposes DyMixOp, a neural operator framework for PDE solving that integrates concepts from complex dynamical systems and inertial manifold theory. The authors aim to provide a “theory-guided” architecture for operator learning by reducing infinite-dimensional nonlinear PDE dynamics to finite-dimensional latent dynamics. The core architectural contribution is the Local-Global-Mixing (LGM) transformation, which combines localized integral operators and global (spectral) operators through element-wise multiplicative interactions. The authors claim this design is inspired by the convection term in fluid dynamics and enables modeling both local and global interactions in a unified manner. DyMixOp stacks multiple LGM layers in a “dynamics-informed” architecture, intended to reflect temporal evolution in dynamical systems. Empirical results on several PDE benchmarks (e.g., 1D KS, 2D Burgers, Darcy, 2D NS, 3D SW) suggest that DyMixOp achieves lower error compared to standard neural operator baselines, while maintaining modest computational cost and scalability.

**Strengths:**

- Clear high-level motivation: The paper identifies a real problem — the gap between heuristic neural operator architectures and the underlying structure of dynamical systems. This is a relevant and timely research direction.

- Systematic architectural framing: The authors attempt to derive their model structure from inertial manifold theory, rather than relying purely on empirical heuristics. This is more principled than many recent operator-learning works.

- Innovative LGM transformation: Combining local convolution-like operators with global spectral components in a multiplicative fashion is an interesting design choice, loosely motivated by physical convection mechanisms.

- Broad experimental coverage: The paper evaluates DyMixOp across different PDE types (parabolic, elliptic, hyperbolic) and dimensions (1D, 2D, 3D). There are also ablation studies and scaling experiments, which are valuable for assessing model robustness.

- Computational considerations: Some effort is made to analyze memory and training time scaling, which is often ignored in PDE learning papers.

**Weaknesses:**

While the paper has clear ambitions, the actual contribution is significantly weaker than its presentation suggests:

- Overstated novelty and vague theoretical link: The claim of “theory-guided” design is not rigorously substantiated. The connection to inertial manifold theory is superficial: the theory motivates a projection but does not dictate the actual LGM architecture. Similar “local + global” designs exist in Fourier Neural Operator, Wavelet neural operator: a neural operator for parametric partial differential equations, and other hybrid models. The multiplicative fusion is not a substantial conceptual leap.

- No rigorous ablation on theoretical claims: The inertial manifold motivation is not tested empirically. There are no experiments isolating the effect of the projection, the impact of manifold dimensionality, or comparisons against standard dimensionality reduction (e.g., PCA or learned linear projections). It’s unclear whether performance gains come from the theoretical foundation or simply from better inductive bias and model capacity.

- Limited methodological novelty: LGM transformation is just a multiplicative combination of a local and a global kernel — conceptually quite close to existing hybrid architectures. The “dynamics-informed architecture” amounts to a residual stack with learnable step sizes, which resembles well-known operator integration schemes.

- Lack of clarity and rigor in analysis: The theoretical part is mostly descriptive, without formal guarantees connecting DyMixOp’s approximation power to the inertial manifold theory. Key details (e.g., effect of ∆t learnable steps, manifold dimensionality) are glossed over.

- Unbalanced experiments: Most benchmarks are relatively standard and well-behaved. There is no evaluation on irregular domains, non-smooth dynamics, or truly stiff PDEs. Performance gains on simpler PDEs are marginal; large reported improvements mainly appear in convection-heavy cases, which may favor the architecture’s bias rather than indicating broad generality.

- Overclaiming in narrative: Statements like “principled foundation” and “direct implementation of mathematical structure” are strong but not justified by the actual level of theoretical rigor or experimental support.

**Questions:**

- Empirical grounding of theory: How exactly does the inertial manifold motivation improve model performance? Please show experiments isolating the projection and manifold dimensionality effects, and compare to naive PCA or linear projections.

- Role of the LGM transformation: What happens if the multiplicative fusion is replaced with additive or concatenation-based fusion? Is the multiplicative interaction critical or just one of many equivalent options?

- Scaling and generalization: How does the method perform on irregular meshes, complex geometries, or PDEs outside the training family? How robust is the architecture to resolution and parameter shifts?

- Theoretical rigor: Can the authors provide formal approximation guarantees or bounds connecting the inertial manifold structure and the expressiveness of LGM layers? Otherwise, the claims of “principled design” are misleading.

- Efficiency trade-offs: Provide explicit runtime and parameter count comparisons for each baseline and DyMixOp. Does the added theoretical structure introduce training overhead?

---

### Official Review · Reviewer_zWnC · 2025-10-27

**Soundness:** 2
**Presentation:** 2
**Contribution:** 2
**Rating:** 2
**Confidence:** 4

**Summary:**

This paper introduces DyMixOp, a neural operator framework for solving PDEs that integrates insights from complex dynamical systems theory. The core contribution they propose is grounded in inertial manifold theory, which enables the transformation of infinite-dimensional nonlinear PDE dynamics into finite-dimensional latent representations while preserving essential non-linear interactions. The authors propose a Local-Global-Mixing (LGM) transformation inspired by convection dynamics in turbulence, which captures both local fine-scale features and global nonlinear interactions through element-wise multiplication of local and global kernel outputs. Further they parameterize the architecture to resemble a forward in time evolution operator with learnable timestep sizes. The benchmarks carried out show advantage of this architecture over baselines on the test problems considered.

**Strengths:**

- A core strength of this paper is its grounding in dynamical systems theory. Instead of proposing a new architecture based purely on heuristics, the authors use inertial manifold theory to justify their model's core components with well motivated design choices. Further, their design on the LGM block based on a mix of local and global dynamics and seems to play out empirically as well with the obtained results on convection dominated problems.
- Shown to demonstrate good performance beating established architectures such as the FNO on the chosen datasets.

**Weaknesses:**

- The paper builds a strong theoretical case for using inertial manifold theory, which involves a global projection onto a low-mode basis. However, the actual implementation of the projection layer ($\mathcal{P}_m$), the dimension-shifting layer ($\mathcal{T}$) and the local component of the mixing blocks all use 1x1 convolutions. A 1x1 convolution would be a local, point-wise operator that mixes channels (effectively just a dense transform IIUC). The authors claim that this isn't necessary with larger kernel sizes being used for more performance-intensive applications but this doesn't seem to have been explored and seems contradictory.

- The experiments report prediction errors, but the time horizon is not specified. For dynamical systems, the most critical metric is often long-term stability and error accumulation in autoregressive rollouts. I would expect that with an architecture motivated by the inertial manifold, the learned dynamics should be consistent even through the rollout.

- Further, the claim is that this architecture alleviates the spectral bias inherent to neural operators isn't well evidenced. I would like to see comparisons depicting that this architecture shows lower error in the higher bucket of frequencies as compared to the standard architecture.

- I'm not entirely convinced that these benchmarks are robust enough, and compare against the current advancements in neural operator design. Can you perhaps adopt a few more standard benchmarks that have established results such as the PDEArena benchmarks / TheWell dataset?

**Questions:**

Please address the weaknesses. Additionally:

- Could you please clarify what is meant by the "combined parallel-hierarchical structure" or "hybrid variant"? From my understanding, the architecture is purely sequential.
- Can you clarify on the choice of your hyperparameters? I see that scaling behaviour increases both the width and the layer count. Given that there's a low manifold dimension that we're hoping to uncover, shouldn't we see a saturation in the performance as we increase the width? Have you looked into this?
- Are all of the benchmarks performed acquired from somewhere or generated? I can see there's citation for 2D Darcy's and 3D SW but not for the others. As mentioned in the weaknesses, can you perhaps do a comparison to established benchmarks such as PDEArena or TheWell?

---

### Official Review · Reviewer_3Beo · 2025-10-30

**Soundness:** 2
**Presentation:** 1
**Contribution:** 1
**Rating:** 2
**Confidence:** 5

**Summary:**

This paper introduces DyMixOp, a neural operator framework guided by inertial manifold theory and complex dynamical systems to learn PDE dynamics efficiently. The method combines local and global transformations through a Local-Global-Mixing (LGM) mechanism, designed to capture both fine-scale and domain-wide interactions.

**Strengths:**

1) Derivation from inertial manifold theory provides a rare, principled approach to neural operator design.

2) The proposed Local-Global-Mixing transformation effectively combines spectral and convolutional representations.

**Weaknesses:**

1) Architectural implementation details are dense and difficult to follow, and may be difficult to reproduce as the codebase is also not provided.

2) Does not benchmark against some recent operator designs (e.g., Latent Mamba Operator, Transolver, GeoFNO).

3) No systematic exploration of how latent space size affects accuracy or stability.

4) While qualitative results on efficiency are provided, quantitative analysis of FLOPs or runtime scaling is missing.

5) The treatment of irregular geometries or mixed boundary conditions remains somewhat unclear.

**Questions:**

1) How sensitive is the performance of DyMixOp to the chosen latent dimension or projection rank?

2) Can the Local-Global-Mixing transformation be adapted to irregular meshes or graph-based domains?

3) How does the framework handle non-smooth or discontinuous boundaries—is the inertial manifold assumption still valid there?

4) Could DyMixOp be extended to time-dependent or multi-physics systems, and how would the architecture scale in such cases?

5) Have the authors analyzed or visualized the learned spectral kernels to validate the claimed physical interpretability?

6) No codebase and hyperparameter details are provided.

**Details Of Ethics Concerns:**

Dual Submission (https://arxiv.org/pdf/2508.13490). Paper submitted to AAAI 26 as well, simultaneously.

---

### Official Review · Reviewer_hWWt · 2025-11-01

**Soundness:** 3
**Presentation:** 3
**Contribution:** 3
**Rating:** 6
**Confidence:** 4

**Summary:**

This paper introduces Dymixop, a neural operator framework inspired by inertial manifold theory from complex dynamical systems. The key idea is the Local-global-mixing transformation, combining the local and global operations via element-wise multiplication. The framework transforms infinite dim PDE dynamics into finite-dimensional ones while preserving nonlinear interactions. Experiments show the efficacy of the method.

**Strengths:**

1. The proposed architecture is grounded in inertial manifold theory.

2. The LGM transformation is conceptually elegant and well motivated.

3. Experiments are fairly comprehensive.

4. Results are good both from performance and efficiency angles.

**Weaknesses:**

1. The connection between the inertial manifold theory and actual implementation is loose. Theory assumes specific decompositions and projections but implementation uses learnable convolutions that does not relate to p_m projections in theory.

2. I think the novelty is overstated, given that elementwise multiplication of local and global features isn't new.

3. Many recent baselines are missing - Transolver, UNO etc, FNO is a little dated.

4. A lot of design choices are unclear.

5. Notational inconsistencies exist throughout the paper.

6. There are no resolution interference, OOD generalization, and computational complexity experiments/analysis.

**Questions:**

1. Can it be proved that learnable 1x1 convolutions and P_m actually implement the theoretical projections from the manifold theory?

2. Are there any guarantees coming from theory for the learned representations?

3. How does the method work when inertial manifolds don't exist for a given PDE?

4. The claim about natural decomposition into local and global features needs to be formalized.

5. Can you provide justifications for the hyperparameter choices made?

6. Can the model handle different spatial resolutions at training and test times?

7. The consistency loss term in Eq. 15 seems like an Auto-encoding loss. How is this specific to the given theory?

9. What are the failure cases of the proposed formulation?

---

### Note · Authors · 2025-11-29

I have read and agree with the venue's withdrawal policy on behalf of myself and my co-authors.